# Pulmonary Metastasising Aneurysmal Fibrous Histiocytoma: A Case Report, Literature Review and Proposal of Standardised Diagnostic Criteria

**DOI:** 10.3390/diseases11030108

**Published:** 2023-08-23

**Authors:** Fiona Mankertz, Rebecca Keßler, Andrea Rau, Christian Seebauer, Silvia Ribback, Alexandra Busemann

**Affiliations:** 1Institute for Diagnostic Radiology and Neuroradiology, University Medicine Greifswald, Ferdinand-Sauerbruch-Str., 17475 Greifswald, Germany; 2Department of Oral and Maxillofacial Surgery/Plastic Operations, University Medicine Greifswald, Ferdinand-Sauerbruch-Str., 17475 Greifswald, Germany; 3Institute of Pathology, University Medicine Greifswald, Friedrich-Loeffler-Straße 23e, 17475 Greifswald, Germany; 4Department of General, Visceral, Thoracic and Vascular Surgery, University Medicine Greifswald, Ferdinand-Sauerbruch-Str., 17475 Greifswald, Germany

**Keywords:** aneurysmal fibrous histiocytoma, AFH, angiomatoid fibrous histiocytoma, dermatofibroma, differential diagnosis

## Abstract

An aneurysmal fibrous histiocytoma is a rare cutaneous soft-tissue tumour which accounts for approximately 0.06% of all dermatopathologies. Metastasis is exceedingly uncommon, to the point that there have only been eight reported cases in the scientific literature. We present the case of a 25-year-old male with a primary aneurysmal fibrous histiocytoma located in the nuchal region which exhibited rapid growth and abrupt ulceration over a short time span and showed signs of locoregional aggressive infiltration. A subsequent histopathological analysis confirmed the presence of diffuse solid and cystic pulmonary metastases. Further genetic sequencing verified LAMTOR1-PRKCD fusion. This case report seeks to review the existing literature on aneurysmal fibrous histiocytoma, discuss the challenges of differential diagnosis and propose standardised diagnostic criteria.

## 1. Introduction

An aneurysmal fibrous histiocytoma is a rare skin tumour comprising only 0.06% of all cutaneous soft-tissue lesions. The term “fibrous histiocytoma” encompasses a group of benign skin tumours characterized by distinct microscopic characteristics rather than their histopathological aetiology [1]. The classification of the fibrous histiocytoma was revised by the World Health Organization (WHO) in 2002 to encompass its multitude of appearances and growth patterns. The most common form is the banal fibrous histiocytoma, which represents over 80% of all fibrous histiocytomas and exhibits distinctive morphological characteristics which can be diagnosed through visual inspection and clinical correlation alone. However, clinical diagnosis proves to be more difficult when examining the more rare subvariants of the fibrous histiocytoma, one of which is the aneurysmal fibrous histiocytoma [2].

The aneurysmal fibrous histiocytoma, first described by Santa Cruz et al. in 1981, comprises less than 2% of all fibrous histiocytomas. Its short tumour-volume doubling time, characteristic livid colouring, high probability of haemorrhaging and comparatively high rate of post-treatment recurrence emulates the growth patterns and morphological characteristics of malignant soft-tissue tumours such as melanoma [3,4,5]. Due to its rarity, an accurate differential diagnosis requires a comprehensive evaluation involving histopathological and immunohistochemical examinations. Additionally, although the common variant of the banal fibrous histiocytoma is non-invasive and considered benign, the aneurysmal fibrous histiocytoma shows signs of aggressive locoregional growth as well as distant metastasis in exceptionally rare cases [2,6,7,8]. Malignant fibrous histiocytomas have been documented to show a wide variety of sites of dissemination such as cardiac, vascular and cerebral metastases [9,10,11].

We present the case of a young male patient with a recurrent nuchal aneurysmal fibrous histiocytoma and confirmed pulmonary metastasis. We describe the patient’s demographics, symptoms, clinical, radiographic and immunohistopathological diagnostics and the course of treatment. Additionally, we compile all the cases of metastatic aneurysmal fibrous histiocytomas between 1981 and 2023. We used the databases PubMed, Cochrane Library and Wiley to search the terms “aneurysmal fibrous histiocytoma” AND “metastasis” or “angiomatoid fibrous histiocytoma” AND “metastasis” or “malignant fibrous histiocytoma” AND “metastasis”, as well as to review the search term “aneurysmal fibrous histiocytoma” in the PubMed database when sorted by date of publication up to the year 2000. We selected a total of ten articles: four articles on aneurysmal fibrous histiocytomas and six on angiomatoid fibrous histiocytomas. We excluded any cases of malignant metastatic fibrous histiocytomas that were not aneurysmal or angiomatoid when histopathologically examined. Additionally, we excluded any cases that did not provide sufficient evidence of metastasis or were not histopathologically examined.

In total, seven metastasised cases of aneurysmal fibrous histiocytoma have been reported at this point in time [6,8,12,13,14]. Ours is the eighth case.

In the discussion, we compare and contrast our case report with the existing literature and elaborate upon the similarities and differences in histopathological and immunohistochemical diagnoses. Additionally, we propose three standardised diagnostic criteria which may aid clinicians in choosing treatment protocols.

## 2. Case Report

A 25-year-old Caucasian male patient with prior dermatopathological history presented to the surgical department of the University Medicine Greifswald with a nuchal soft tissue mass on the left side of the trapezius muscle. At the time of admission, the patient reported a height of approximately 180 cm (5′10″) and a weight of approximately 82 kg (approximately 181 lbs). His occupation was in the culinary field. The patient reported that he had first noticed the mass approximately 14 months ago, when it was externally brought to his attention. Over this time period, he observed a gradual, steady increase of the mass’s size without abrupt changes in its volume, colour or consistency. This gradual increase remained stable up until three weeks before his initial presentation to the surgical department. Throughout the course of those three weeks, the mass showed a sudden, accelerated growth and a subjective darkening of the cutaneous surface, which led to his initial presentation.

The sole symptom reported by our patient was a dull, steady pain upon palpating the mass. He negated symptoms such as sudden weight loss, fever, night sweats or dyspnoea. The patient’s dermatopathological history included a prior diagnosis of aneurysmal fibrous histiocytoma at nine years of age. This tumour had presented as an interscapular livid soft tissue mass measuring 20 × 40 × 5 mm which was initially clinically diagnosed as a benign haemangioma by the paediatric department of the University Medicine Greifswald. Further histopathological testing resulted in a final diagnosis of aneurysmal fibrous histiocytoma. A surgical resection of the aneurysmal fibrous histiocytoma was performed after the histopathological diagnosis. In the post-surgical histopathological report examining the resected tissue, it was noted that the tumourous tissue extended closely towards the border of the deep resection. Due to the close resection border and the relatively high potential for recurrent disease, which is 20–30% in the case of an aneurysmal fibrous histiocytoma, regular follow-ups were recommended by the paediatric oncological department of the University Medicine Greifswald [15]. Those follow-ups were not complied with due to unknown reasons. When examining the family history for cases of cutaneous soft-tissue tumours or other dermatopathologies, we found no familiar history of such aside from a diagnosis of psoriasis received by a maternal grandparent. One paternal grandparent had passed away from aggressive thyroid cancer.

Upon initial clinical examination, we saw a soft tissue mass consisting of two livid, dome-shaped extrusions with dimensions of about 30 × 20 × 20 mm (left) and 20 × 20 × 20 mm (right). These two masses were connected in the middle and showed a disseminating brown-red border of a diameter of about 5–10 mm in all directions. The mass showed an intact skin barrier. It did not exhibit any ulcerations or necroses at this point in time, although the centre of the right mass showed partial squamous skin flaking (Figure 1). Upon a light touch, the patient reported the intact sensibility of the skin nerves. Light palpation was not painful, whereas firmly palpating the centre of the right mass was painful for the patient.

The patient was admitted to the surgical ward for further diagnostics. Laboratory values for venous blood were all in the normal range. Following a multidisciplinary board conference with the department of dermatology and the department of radiology, a soft-tissue CT scan of the neck was recommended to determine the nuchal growth, the width of infiltration and the presence of any local or disseminated metastases.

The initial CT scan of the soft-tissue in the neck scan showed a locally invasive 52 × 39 × 25 mm soft-tissue mass of subcutaneous tissue. Two independent radiologists documented the infiltration of the ipsilateral M. trapezius through multiple disseminating and invasive local satellite lesions. Additionally, multiple soft-tissue dense lesions in the ipsilateral M. pectoralis major and the M. psoas major were documented. In line with standardised diagnostical protocol, a punch biopsy of the nuchal mass was obtained through the Institute of Diagnostic Radiology, Greifswald, and evaluated through the Institute of Pathology, Greifswald.

A histopathological evaluation of the punch biopsy showed spindle cells in a swirling formation with hemosiderin-laden macrophages and lipid-laden macrophages, as well as the infiltration of fascia-like collagen bundles (Figure 2a). Limited cell mitosis was observed, whereas tumour necrosis was not. The tumour cells partially tested positive for CD34 as well as CD68 but tested negative for beta-Catenin, Desmin, S100-Protein or STAT6 (Figure 2b). In additional tests, no expression of AE1/3, Caldesmon and Zytokeratin was shown. Proliferation activity (Ki-67-Index) was consistently reported to measure around 10%. At this point in time, a definite diagnosis was unclear. As such, a targeted sequencing assay (FusionPlex Sarcoma v2 Panel) for 63 gene fusions associated with soft-tissue cancers was performed. The tissue exhibited a fusion of LAMTOR1-PRKCD Breakpoint chr11:71814229, chr:53219620. This specific fusion has been correlated with the presence of aneurysmal fibrous histiocytoma, which will be further discussed in the literature review and discussion [16].

As a subsidiary finding of the initial nuchal CT scan, multiple dense soft-tissue structures were observed in the apical pulmonary tissue of both lung sides. This raised concerns for possible malignancy as well as the presence of metastatic disease. In line with standardised follow-up staging protocols, as confirmed by a multidisciplinary board conference, a staging CT scan of the thorax and abdomen was conducted. In this scan, a multitude of pulmonary masses were documented: we saw bilobar masses of both hyperdense solid (n > 30) and hypodense cystic (n > 40) consistencies. These masses are shown in Figure 3, with the solid masses circled by a yellow border and the cystic masses circled by a red border. Some of the solid masses were septally divided, and several solid masses were directly adherent to the cystic lesions. The largest solid lesion in the left lower pulmonary lobe measured approximately 21 mm (Figure 3a), whereas the largest cystic lesion, located in the left upper pulmonary lobe, measured approximately 35 mm (Figure 3b).

The radiological department conducted a second successful punch biopsy of the largest solid mass in the left lower pulmonary lobe to discern the presence of metastatic disease.

Upon histopathological evaluation, a solid mass composed of spindle cells mixed with few round cells was reported (Figure 4a). In immunohistochemical staining, the spindle cells partially tested positive for the expression of CD34 and weakly positive for smooth muscle actin (Figure 4b). Significant histopathological similarities between the nuchal tumour and the pulmonary mass were confirmed to the point of a unilateral agreement on the presence of pulmonary metastasis. Due to the nuchal tumour’s recent aggressive growth, a multidisciplinary board conference extensively discussed treatment options. Surgical excision of the nuchal tumour was recommended. However, the patient chose a watch-and-wait approach due to reporting a subjective lack of symptoms at this point in time. In an initial follow-up by the surgical department a month after the multidisciplinary board conference, the nuchal tumour was stable and constant in size.

However, approximately four months after undergoing the initial testing, the patient reported to the surgical department of the University Medicine Greifswald again. This time, he reported an abrupt increase in pain upon palpation as well as bloody discharge from the nuchal mass. He could not succinctly provide details as to the time frame over which the abrupt increase and the cutaneous discharge occurred, but he conferred a time span from approximately ten days to two weeks. Upon clinical assessment, we observed a significant increase in size, from an initial horizontal length of 50 mm to 82 mm, as well as an almost complete ulceration of the epidermal layer (Figure 5a). At an immediate multidisciplinary board conference, complete resection of the tumour and tissue reconstruction with an ALT (anterolateral thigh) microvascular free flap were recommended. This was successfully completed through a joint operation conducted by the Department of General, Visceral, Thoracic and Vascular Surgery and the Department of Oral and Maxillofacial Surgery.

According to standardised protocols, the nuchal tumour was submitted for histopathological and immunohistochemical evaluations. A histopathological examination of the ulcerated mass showed a nodular spindle-cell tumour with large, hyperchromatic nuclei. This was identical to the initial histopathological evaluation performed when the patient had first presented four months earlier. The tumour had infiltrated the subcutis and the trapezius muscle, as well as smaller venous tissue directly adjacent to the tumour. The resection was completed in toto with tumour-free borders. At a 2-week post-operative appointment, the patient reported no further complications. Physiological wound healing in the nuchal region was observed, with no signs of perfusion problems for the ALT free flap and textbook skin-colour matching of the flap and the surrounding skin (Figure 5b). The patient reported no dysesthesia or localised pain.

Treatment options for the pulmonal solid lesions were extensively discussed by the general surgical department, the oral and maxillofacial surgical department, the pathological department the radiological department and the oncological department in a multidisciplinary board conference. Multiple-surgical-excision treatment, local pulmonary ablation through radiofrequency ablation or microwave ablation and systemic chemotherapy or radiation therapy were discussed. The patient showed an explicit inclination towards a watch-and-wait therapy and opted instead for further observation of both the nuchal region of the primary tumour and the pulmonary metastases. A 6-month follow-up post-surgery showed no progress of the pulmonal lesions, while a 9-month follow-up showed no local recurrence of the nuchal tumour. As such, at this point in time, the patient is considered alive with disease.

Before the nuchal surgery, the patient consented to written, photographic and radiographic documentation and publication of his case. He was extensively informed about his demographic information that might be published and of the possibility of publication in a medical journal. Written consent was provided and witnessed.

## 3. Literature Review

We conducted a review of the literature on both metastatic aneurysmal fibrous histiocytomas as well as angiomatoid fibrous histiocytomas (Table 1). This was complicated by the partially overlapping definitions of the terms “aneurysmal” and “angiomatoid” when referring to a fibrous histiocytoma. While Santa Cruz and McKenna used both aneurysmal and angiomatoid fibrous histiocytomas synonymously in 1981 and 1999, Zelger refers to angiomatoid fibrous histiocytomas as deeper, slightly more aggressive counterparts to aneurysmal fibrous histiocytomas [5,17]. In newer research, Wood differentiates angiomatoid fibrous histiocytomas via several histopathological and immunohistochemical characteristics such as “the presence of peripheral collagen trapping and lack of a complete fibrous pseudocapsule, peripheral lymphoplasmacytic inflammation and angiomatoid fibrous histiocytoma associated translocations”, such as a typical rearrangement of or partial fusion of the EWSR1 [6]. Antonescu et al. defined EWSR1-CREB1 fusion as a pathognomonic feature of angiomatoid fibrous histiocytomas [18]. For the purpose of providing a full overview of metastatic fibrous histiocytomas, we chose to include angiomatoid fibrous histiocytomas as an additional subtype but will not be comparing this subtype to our case report.

Several similarities between the cases of metastasising aneurysmal fibrous histiocytomas were noted. A comparison and contrast with our case report will be provided in the discussion.

All aneurysmal fibrous histiocytomas in the reported cases (n = 7) first presented as indolent, livid masses in the subcutaneous tissues of the extremities or the dorsal trunk. Macroscopically smooth, dome- or oval-shaped borders and an intact skin barrier were documented. In n = 3 cases, the patients reported an increase in pain upon palpation. A longer stretch of continuous, slow increase in size followed the initial presentation. In all cases except one, the soft tissue mass abruptly increased in size in a span of days to weeks and the skin barrier was disrupted and was accompanied by ulceration, bleeding and an increase in subjective pain or discomfort. In all reported cases but one, time to metastasis was recorded between 6–32 months. In all reported cases, at least one lymph node was histopathologically confirmed as the site of metastasis (n = 7). The second most common site of metastasis was pulmonary tissue (n = 5) [6,8,12,13,14].

Histopathological examinations showed a common pattern of swirling fibroblasts and haemosiderin-laden macrophages, as well as erythrocyte-filled spaces lined by single-layer epithelium. Immunohistochemical testing was consistently positive for CD68 as well as negative for CD34. In cases in which smooth muscle actin was tested it was also positive. At the respective times of publication, n = 4 patients presented with disease-free survival, n = 2 patients were alive with stable disease and n = 1 patient was lost to follow-up [6,8,12,13,14].

## 4. Discussion

We reported a case of a nuchal aneurysmal fibrous histiocytoma with pulmonary metastasis. Despite fibrous histiocytomas accounting for approximately 3% of all presented dermatopathologies, the majority of those cases (>80%) are diagnosed as banal fibrous histiocytomas. Conversely, the aneurysmal fibrous histiocytoma is a very rare variant which constitutes less than 2% of all fibrous histiocytomas. Only a limited number of cases have been recorded in the scientific literature since its initial description by Santa Cruz et al. in 1981, with a total incidence of approximately 0.06% [1,4,17]. In 1995, Calonje et al. reported 40 diagnosed cases in the time span between 1939 and 1995. They attributed the scarcity of the disease in both diagnosis and in the literature to the rarity of the disease as well as the difficulty of clinical diagnosis [12]. Additionally, our patient presented with both cystic and solid pulmonary metastases. Despite its comparatively high recurrence rate, the metastasis of a fibrous histiocytoma has only occurred in exceptionally rare cases. As such, the histopathological examination of these metastatic lesions is required to rule out another malignant primary tumour or further localised disease.

Due to its atypical livid colouring, dolency and atypical fast growth, initial clinical diagnosis might pose a multitude of differential diagnoses. Common differential diagnoses include an angiomatoid fibrous histiocytoma, Kaposi sarcoma, malignant melanoma, atypical angiosarcoma or extraosseous bone sarcoma, such as an Ewing sarcoma [20,23,24,25]. A mere visual diagnosis is often not sufficient, and further histopathological and immunohistochemical testing is required. In at least one case, an aneurysmal fibrous histiocytoma was shown mimic the dermoscopic characteristics of Kaposi sarcoma and malignant melanoma [3,26]. However, even immunohistochemical testing may sometimes prove to be misleading. Sheehan et al. noted the difference in immunoreactivity between a Kaposi sarcoma and aneurysmal fibrous histiocytoma [27]. However, in our patient’s initial histopathological examination, the tumour tested weakly positive for CD34 but negative for S-100. This is an aberration from the results of our literature review, in which all cases in which CD34 was immunohistochemically tested were negative [6,8,12,13,14]. Further testing for HHV-8 and CD68 ruled out a Kaposi sarcoma. The storiform organisation of the cells and the lack of undifferentiated necrosis zones similarly ruled out an angiosarcoma and malignant melanoma. Mitotic activity was low; however Kadda et al. noted that there is no correlation between a high level of mitotic activity and the likelihood of metastasis; conversely, many fibrous histiocytomas that showed high levels of mitotic activity never exhibited recurrence or metastasis [28].

Our patient’s case is unique as in that this is the first recorded case of an aneurysmal fibrous histiocytoma’s primary pulmonary metastasis when compared to thes cientific literature. In all other recorded cases (n = 7), the first sites of metastatic dissemination were locoregional lymph nodes [6,8,12,13,14]. However, our case also shares many similarities with the existing literature. Doyle et al. reported a singular cystic pulmonary metastasis measuring 50 mm in diameter, whereas our patient’s largest cystic pulmonary metastasis measured 35 mm in diameter (Figure 3a). A histopathological examination of our patient’s pulmonary metastases reported that they were morphologically similar to the primary nuchal tumour, with large, haemosiderin-laden and blood-filled spaces as well as CD68-positive fibroblastic cells and sparse mitotic activity. This matched Doyle et al.’s and Wood et al.’s histopathological and immunohistochemical results [6,8].

Upon further reviewing the existing literature, we found the case of a 19-year-old female with an occipitonuchal and buccal aneurysmatic fibrous histiocytoma. This case also displayed the typical signs of aggressive localised disease: ulceration, recent, rapid growth and locally invasive infiltration. LAMTOR1-PRKCD gene fusion was present. Our patient also presented LAMTOR1-PRKCD gene fusion. Panagopoulos et al. proposed that the presence of this chromosomal fusion might be pathognomonic for aggressive soft-tissue tumours. However, it differs from the diagnosis of an angiomatoid fibrous histiocytoma in that LAMTOR1-PRKCD gene fusion is not considered pathognomonic for an aneurysmal fibrous histiocytoma the way EWSR1-CREB1 gene fusion is considered pathognomonic for an angiomatoid fibrous histiocytoma. Jedrych et al. relate this to the comparatively fewer reported metastasising aneurysmal fibrous histiocytomas and propose the further examination of gene fusion in such cases [7,15,16,18,20,21,23].

Our case report is limited in that post-surgical observation has only progressed for approximately twelve months. In follow-up CT scans, the pulmonary metastases were stable in size and dissemination. Further monitoring is needed to evaluate the progress of the disease. Additionally, as the patient declined further therapeutic intervention, further diagnostics are limited. According to our literature review, the surgical treatment of pulmonary metastases is not necessarily required, however [8]. Finally, due to the comparatively miniscule number of case reports, there have not been any attempts to develop standardised diagnostic and therapeutic criteria. However, our case report is salient in that it is the first case of a metastatic aneurysmal fibrous histiocytoma that is CD34-positive and does not report primary locoregionary lymph node infiltration. As such, it might be used as a basis for further differential diagnoses of CD34-positive subcutaneous malignancies.

The nuchal tumour presented by our patient showed three signs of local aggressive disease: (A) the abrupt ulceration of the epidermis; (B) recent accelerated growth after a period of stable disease; (C) chromosomal translocation. In the reported cases of metastasising aneurysmal and angiomatoid fibrous histiocytomas, similar criteria were documented. The most common symptom of an aggressive aneurysmal fibrous histiocytoma is an abrupt increase in the tumour size of the initial lesion. Haemorrhage and an ulcerated cutaneous layer were infrequently reported [4,6,7,8,13,15,19,20,23,29]. However, as of yet, there has not been a comprehensive review attempting to provide standardised diagnostic clinical markers. Based on the evidence provided by the literature review, we wish to recommend these three criteria as markers as a tool for the clinical diagnosis of an aggressive aneurysmal fibrous histiocytoma.

## 5. Conclusions

An aneurysmal fibrous histiocytoma is an uncommon variant of the fibrous histiocytoma which constitutes less than 2% of all fibrous histiocytomas. At this point in time, only seven cases of metastatic aneurysmal fibrous histiocytomas have been recorded [6,8,12,13,14]. A 25-year-old male patient initially presented with an indolent nuchal mass which had developed over a year. Over a subsequent time period of four months, the mass showed a significant increase in growth, as well as ulceration and bloody discharge. The patient was then referred for a surgical excision and microvascular free-flap reconstruction, which were successfully completed. Uniquely, this is the first reported case of a primary pulmonary without the presence of locoregionary lymph node metastasis. At time of publication, the patient is stable with disease.

In a comprehensive review of the literature, a number of similarities, such as macroscopic features, signs of aggressive disease, as well as histopathological configuration, were evident. However, some notable differences, such as the immunohistochemical reactivity of CD34, as well as the lack of lymph node metastasis, were also noted. As such, this case report could prove to be relevant for the further diagnostic advancement and exclusion of differential diagnosis [6,8,12,13,14].

Due to the rare occurrence of aneurysmal fibrous histiocytomas, they are is often misdiagnosed, and no official clinical guidelines for their diagnosis and treatment exist. Extensive histopathological and immunohistochemical evaluations are required to exclude differential diagnoses such as angiomatoid fibrous histiocytoma, Kaposi sarcoma, malignant melanoma, atypical angiosarcoma or extraosseous bone sarcoma such as Ewing sarcoma [3,20,23,24,30,31]. The authors propose three criteria which may indicate the presence of an aggressive aneurysmal fibrous histiocytoma: (A) the abrupt ulceration and haemorrhaging of the epidermis; (B) recent accelerated growth following a period of stable disease or slow growth; (C) the presence of chromosomal fusion or rearrangement in immunohistochemical examination.

We conclude that due to the rarity of this disease, the further development of standardised diagnostic protocols is warranted, which can only be achieved through the continued comparing and contrasting of published cases.

## Figures and Tables

**Figure 1 diseases-11-00108-f001:**
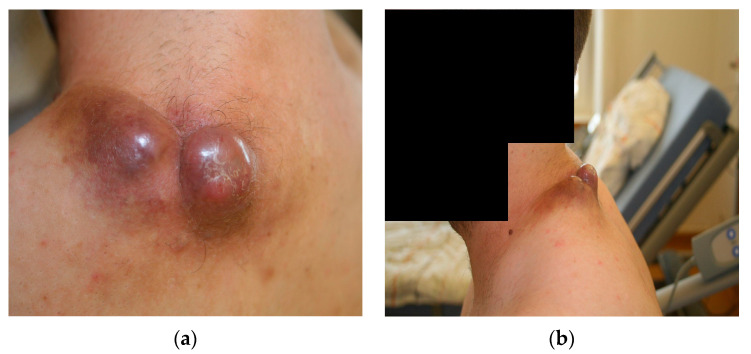
Macroscopic imaging of the initial nuchal mass: (**a**) dorsal projection; (**b**) lateral projection.

**Figure 2 diseases-11-00108-f002:**
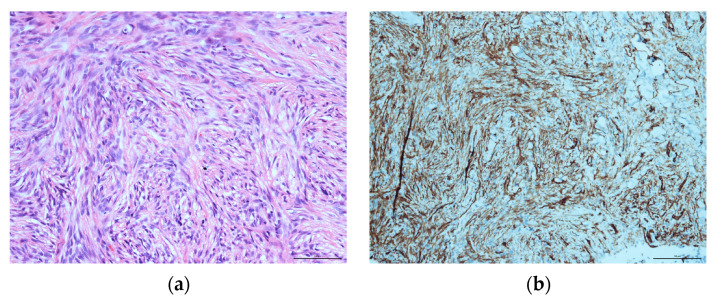
Histopathological evaluation of the initial nuchal mass: (**a**) HE staining, showing swirling spindle cells accompanied by fascia-like collagen bundles at ×400; (**b**) CD34-positive immunohistochemical staining.

**Figure 3 diseases-11-00108-f003:**
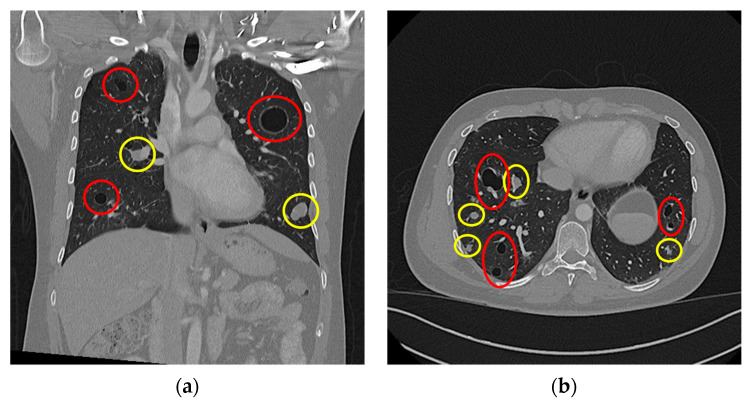
Thoracal CT scan: (**a**) coronar projection; (**b**) axial projection. Both images show multiple cystic (highlighted in red) and solid (highlighted in yellow) lesions.

**Figure 4 diseases-11-00108-f004:**
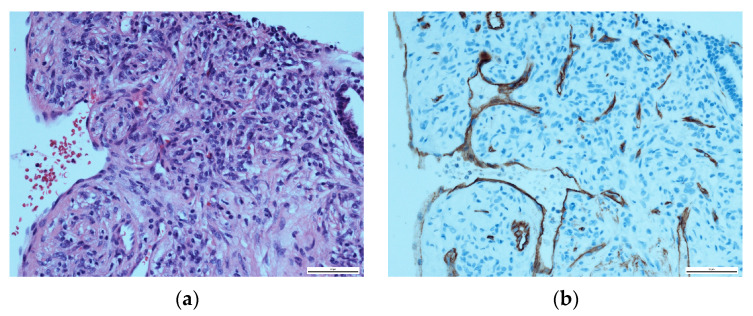
Histopathological evaluation of the solid pulmonary mass: (**a**) HE staining, showing a majority of spindle cells with a few round cells at ×400; (**b**) CD34-positive immunohistochemical staining of the spindle cells.

**Figure 5 diseases-11-00108-f005:**
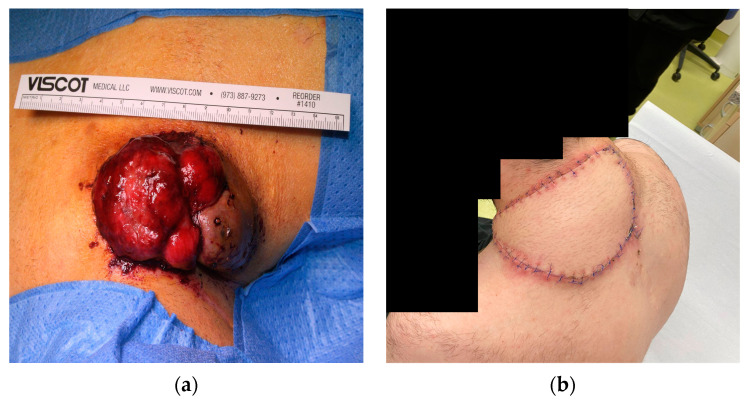
Macroscopic imaging of the ulcerated nuchal mass (**a**) pre-surgery; (**b**) two weeks post-surgery.

**Table 1 diseases-11-00108-t001:** A literature review of metastatic aneurysmal and metastatic angiomatoid fibrous histiocytomas reported during 2000–2023.

Author	Year of Publication	Age [y] (Gender)	Subtype	Primary Site	Metastatic Site	Time to Metastasis (Months)	Molecular Characteristics
Guillou et al. [13]	2000	27 M	Aneurysmal	Right thigh, soft tissue	Right inguinal multinodular lymph nodes	228	CD34 negative CD68 positive
Matsumura et al. [19]	2010	10 M	Angiomatoid	Dorsal soft tissue (unspecified)	Lung (unspecified)	192	CD34 negative CD68 positive ESWR1 rearrangement
Matsumura et al. [19]	2010	54 M	Angiomatoid	Thigh, muscular tissue (unspecified)	Lung (unspecified)	5	CD34 negative CD68 positive ESWR1 rearrangement
Thway et al. [20]	2012	8 M	Angiomatoid	Left posterior scalp, soft tissue	Right postauricular lymph node	36	CD34 negative CD68 positive EWSR1-CREB1 gene fusion
Doyle et al. [8]	2013	42 M	Aneurysmal	Neck (unspecified)	Cervical lymph node Cervical soft tissue Cystic pulmonary lesion	24	CD34 negative CD68 N/A
Doyle et al. [8]	2013	41 M	Aneurysmal	Shoulder (unspecified)	Axillary lymph nodes Bilateral pulmonary nodules	17	CD34 negative CD68 N/A
Mentzel et al. [14]	2013	2 F	Aneurysmal	Right temple, soft tissue	Multinodular cervical and intraparotid lymph nodes	96	CD34 negative CD68 N/A Gains of chromosome 3, 8, 11
Mentzel et al. [14]	2013	26 F	Aneurysmal	Left thigh, soft tissue	Left inguinal lymph node	96	CD34 negative CD68 N/A
Mentzel et al. [14]	2013	35 M	Aneurysmal	Right foot, soft tissue	Bilateral pulmonary nodules Right inguinal lymph node	32	CD34 negative CD68 N/A
Bohman et al. [21]	2014	36 M	Angiomatoid	Thumb (unspecified)	Regional lymph nodes (unspecified)	24	CD34 N/A CD68 positive ESWR1 rearrangement
Maher et al. [22]	2015	11 F	Angiomatoid	Right thigh, soft tissue	Right pelvic soft tissue	18	CD34 N/A CD68 N/A EWSR1-CREB1 gene fusion
Saito et al. [23]	2017	8 F	Angiomatoid	Crural soft tissue	Regional lymph nodes (unspecified)	6	EWSR1-CREB1 gene fusion
Saito et al. [23]	2017	36 M	Angiomatoid	Right intermuscular popliteal tissue	Lung (unspecified)	18	EWSR1-CREB1 gene fusion
Wood et al. [6]	2020	20 F	Aneurysmal	Left shoulder soft tissue	Left scapular soft tissue, satellite nodule Left axillary lymph node	6	CD10 positive CD34 negative CD68 positive
Cazzato et al. [7]	2022	62 M	Angiomatoid	Right arm soft tissue	Right upper pulmonary lobe	18	CD34 negative CD68 positive EWSR1-CREB1 gene fusion
Current Case	2023	25 M	Aneurysmal	Dorsal nuchal soft tissue	Nuchal soft tissue, satellite nodules Pulmonary cystic lesions Pulmonary solid lesions	16	CD34 positive CD68 positive LAMTOR1-PRKCD gene fusion

N/A = not applicable as information was not provided; M = male; F = female.

## Data Availability

Not applicable.

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
