# Peer review of "Pulmonary Metastasising Aneurysmal Fibrous Histiocytoma: A Case Report, Literature Review and Proposal of Standardised Diagnostic Criteria"

_diseases, 2023, doi:10.3390/diseases11030108_

Round 1

Reviewer 1 Report

General comments

In the manuscript by Mankertz F et al, the authors presented a rare case report regarding the metastasizing aneurismal fibrous histiocytoma (AFH), incidentally making some discussion surrounding it. On the whole, the case report was well articulated and well described that has referential value for pathologists working with AFH. However, as it stands, the paper is far from being publishable in that there were still some problems that need to be addressed. The questions were jotted down blow:

Questions:

1.    From my point of view, the current title leaves room to be desired. For example, could the title be changed into case report and literature review?

2.    In the case of content, only one case report seems to be lean. Could the authors include mini-review in regard to AFH in addition to presentation of the case report?

3.    In introduction section, please describe the strategy of the literature review and provide search terms;

4.    A couple of literatures regarding metastasizing AFH, such as PMID: 32394451, PMID: 10874670, PMID: 8310386, PMID: 22618989,PMID: 2558638;

5.    Please Justify the merit of the case report by using the literature review;

6.    In figure 2, in order to better interpret the CT scanning image, it would be desirable to indicate the lesion the authors wanted to highlight using arrowhead or other appropriate symbols with bright color? You know, not every reader even medical practicer has an inkling of how to interpret the CT scanning;

7.    In patient presentation, the patient demographics (age, sex, height, weight, race, occupation) should be provided;

8.    Please photographs of histopathology in addition to CT scanning and skin manifestations as they relate to the case;

9.    There was no mention that permission should be obtained from the patient to use the patient’s photographs or follow institutional guidelines.

10. Please make sure that the patient case presentation provides enough detail for the reader to establish the case’s validity.

11. In discussion part, please compare and contrast the nuances of the case report with the literature review available or similar to yours.

12. In discussion part, please explain or justify the similarities and differences between the case report and the literature.

13. In discussion part, please list the limitations of the case report and describe their relevance.

14. In discussion part, please summarize the salient features of the case report and justify the uniqueness of the case.

15. In discussion section please provide evidence-based recommendations.

Reviewer 2 Report

This is an interesting report, however some minor changes are needed:

-in the case report please specify better the treatment for lung metastases

- in the discussion I reccomend to do a table with a review of the literature adding in the table the most importat variables that you will find in literature. This may increase the interest of your article and the relative citation.

- after performing these changes you can change also the title, specifying that you performed also a literature review.

- please in the discussion include also the clinical and above all the pathological differential diagnoses.
